# Towards Semantic Digital Twins for Social Networks

Rafael Berlanga[1], Lledó Museros[2], Dolores M. Llidó[1], Ismael Sanz[2], and
María J. Aramburu[2]

[1] Dep. de Llenguatges i Sistemes Informàtics, Universitat Jaume I, Spain
berlanga@uji.es, dllido@uji.es
[2] Dep. de Enginyeria i Ciència dels Computadors, Universitat Jaume I, Spain
museros@uji.es, isanz@uji.es, aramburu@uji.es

**Abstract.** This position paper proposes a platform for the creation of
digital twins for social networks as semantic digital twins for people.
These are mainly aimed at simulating human behavior from a cognitive point of view. The proposal relies on a semantic data infrastructure
aimed at analytical purposes, which is directly fed with real data from
social networks. Summarized data and data generation methods are then
combined to produce new data streams according to the analyst requirements. All these data are stored in a dynamic knowledge graph, which
plays a central role in the design of the digital twins. First experiments
will be conducted on two scenarios where semantic data is already available, namely: Tourism and Fashion.

**Keywords:** Digital Twins · Data Generation · Knowledge Graphs.

## 1 Introduction

Digital Twins (DT) [4] can be defined as (physical and/or virtual) machines or
computer-based models that are simulating, emulating, mirroring, or "twinning"
the life of a physical entity, which may be an object, a process, a human, or a
human-related feature. DTs allow simulations and what-if models of analysis
in order to optimize resources and processes that would otherwise take a long
time to implement in the real environment. The main requirement of a DT is
the massive collection of data from the physical environment to build models
and algorithms that emulate its behavior. In the case of a business environment,
DTs can be created thanks to the digitization of companies, and these DTs are
able to emulate the processes of the company and optimize them in the best
possible way. Artificial Intelligence (AI) also plays a very important role both in
the creation of the DT and its subsequent predictive analysis. To support their
creation, AI techniques such as generative learning models and cognitive models
of the people who take part of the organization are key for a correct definition of
a DT. For predictive analysis, the modeling of data streams and the analysis of
the time series generated by the DT are the main approach to decision making
and the optimization of the processes involved.

DTs are starting to be general-purpose tools, but the adoption of AI and DT is hardly visible these days. In fact, Gartner market research predicted that by 2022 [5], more than two-thirds of companies that have implemented IoT will have deployed at least one DT in production. At the moment, DTs are widely used in manufacturing to optimize asset performance, improve process efficiency, and minimize time and costs. DT is also increasingly finding applications in health care [6], construction [7] and smart cities [8]. Extrapolating smart capabilities from a DT approach to other sectors is challenging, not only from an implementation perspective but also from an ethical and social point of view.

In this position paper, we present our approach for developing a platform that enables the definition of DTs from cognitive and social network data as semantic digital twins for people. The main aim of the intended DTs is to simulate human behavior from a cognitive/social point of view. More specifically, these DTs will rely on AI cognitive models which will be derived from social networks data. As a result, we will be able to simulate different situations and analyze the effectiveness of decisions taken (what-if analysis). The proposed platform will be applied to two verticals, namely: Tourism and Fashions.

## 2   Related Work

In this section recent work related to the development of DTs and Cognitive DTs are presented, including DTs for social media. The proposal of this paper will follow this last trend, developing a Social and Cognitive DT. For developing the new cognitive and social digital twins it is necessary to capture and analyse data from different platforms and sources which might be heterogeneous in syntax, schema, or semantics, making data integration difficult. Therefore, the new DT will make use of the dynamic SLOD-BI platform [1] for capturing and analysing the social data needed for the DT construction. An overview of this platform is also introduced in this section. Finally, for the DT creation generative models are needed, therefore recent work on them are also introduced.

### 2.1   Cognitive Digital Twins

The paper [13] defined a *Cognitive Digital Twin* as a "digital representation, augmentation, and intelligent companion of its physical twin as a whole, including its subsystems and across all of its life cycles and evolution phases". Cognitive Digital Physical Twins (CDPT) will continue optimizing their cognitive, digital and physical design and capabilities over time based on the data they will collect and the experience they will gain, not only based on models and data we gave to them or they inherited. Cognitive Digital Twins will have the abilities of physical and digital self-diagnostic and self-healing systems. CDPT will use different techniques to extrapolate and generate their own version of the reality based on parameters and rules such as time, experience, context, situation, and self and/or environmental awareness - machine perception.

In [10], the challenges of the Cognitive Digital Twins for the Process Industry are presented. In [9] the authors propose an architecture for the implementation of Hybrid (HT) and Cognitive Twins (CT). A CT is a hybrid, self-learning, and proactive system that will optimize its own cognitive capabilities over time based on the data it will collect and experience it will gain.

[20] focuses on DT for reproducing human cognitive processes in cyber-simulation. They define Cognition DT as a model that monitors, and predicts a person's cognitive status through the processing of different type of information.

Finally, on the context of social media, [11] the DT paradigm has been considered to establish a link among social media data analysis for a virtual product. Being able to know the level of intensity of the sentiment of customers for a new product gives higher confidence to the companies and firms when designing a product. This research has attempted to use AI tools to categorize the sentiment trends and fill the gap for the relationship between user emotions and product design. Most of the research on Social Media has been focused on developing algorithmic methods using data-driven approaches. In [12], authors propose PHONY, an automatic system for creating fake news datasets suitable for machine learning algorithms.

## 2.2   Dynamic SLOD-BI

The main motivation behind SLOD-BI (Sentiment Linked Open Data for Business Intelligence) was to build a data infrastructure aimed at sharing extracted sentiment data from social networks [2]. SLOD-BI provides the necessary vocabularies and ontologies to express social network data as well as the analysis patterns for business intelligence (BI) tasks. For example, the concept $UserFact$ accounts for all the observed facts around user accounts, regarding its metrics (e.g. followers), their interactions with other users, as well as their inferred profiles. $SocialFact$ regards the sentiment data generated by these users with respect to some product/service described in the infrastructure. SLOD-BI datasets are intended to cover distinct vertical domains (e.g., automotive, medicine, etc.) so that the corresponding community can fetch queries, gather analytical data and perform analytical queries.

The main drawback of SLOD-BI is that it focused on generating static datasets like other LOD projects. However, social networks are extremely dynamic which are not well suited for LOD nor BI tools. Instead, social data must be regarded as a continuous stream where dimensions continuously change. Thus, we proposed Dynamic SLOD-BI [1], where every element was modelled as a stream. In this scenario, semantic data is stored in a knowledge graph (KG) that is continuously updated. Fig. 1 shows the main entities and BI patterns proposed for (Dynamic) SLOD-BI.

In this paper, the main goal is adapting this infrastructure in order to build digital twins of social network streams. More specifically, we aim at re-using semantic and summarized data from SLOD-BI to simulate new data streams coping with some specific constraints and parameters.

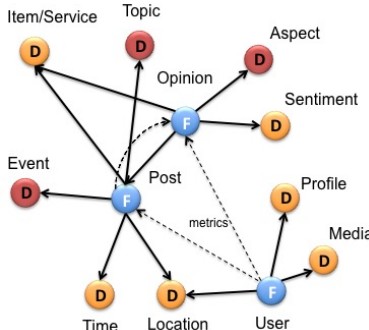

**Fig. 1.** Dynamic SLOD-BI patterns: $F$ represent facts, $D$ represent dimensions (dynamic ones in red). Fact metrics are not showed in the figure.

### 2.3   Generative Models

Generative models are machine learning methods that estimate the join distribution of target and training data. This learned distribution can be used to generate new data similar to the data the models were trained on. The current methods have matured to the point in which they are able to generate very high quality data, text and images. They have produced many practical applications, including highly visible ones such as the generation of photo-realistic images, and they have also been used for non-image data such as time series [15].

Current techniques are almost universally based on Deep Learning. Neural approaches for the development of generative models including Deep Belief Networks Boltzmann Machines, Variational Autoencoders and transformers, which are used for text generation [14]. A particularly important class of methods are *Generative Adversarial Networks*, or GANs, which are based on the interplay between two neural networks (a *generator* which generates candidate data, and a *discriminator* which evaluates it). This technique is behind many currently state-of-the-art results. Deep generative models are starting to be recognized as a relevant tool for the construction of Digital Twins. In recent works they have been applied in an industrial context [17] and also to COVID-19 pandemic modeling [16].

The main challenge in this project is how to adapt existing techniques to the specific needs of a social network DT.

## 3   Overview of the proposal

Fig. 2 depicts an overview of our proposal. Basically, it consists of three main parts, namely: (1) the knowledge graph, (2) the data stream generator, and (3) the analytical and predictive tools at which the generated data is aimed at. These are explained in turn.

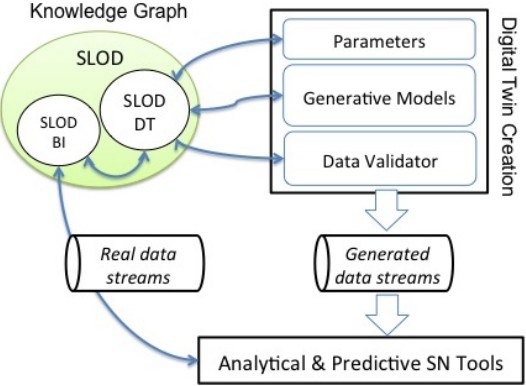

**Fig. 2.** Overview of the proposed architecture for developing Social Network DTs.

### 3.1 The Knowledge Graph

The knowledge graph (KG) includes all existing vocabularies for SLOD-BI as well the new required vocabularies for designing DTs. Parameters and probabilistic distributions will be taken from the SLOD-BI infrastructure, since they regard the elements analysts targeted to. Gathered streamed data in SLOD-BI will serve as a basis for estimating all required distributions that will guide the generation of the new data streams. The KG must be extended in order to regard DT generative methods and how they are linked to SLOD-BI concepts. This will allow designers to choose the most appropriate generative methods depending on the parameter settings. We will adopt a similar approach to that of BigOWL [3] where machine learning methods are represented as semantic data in order to choose the most appropriate methods in a specific Big Data scenario.

### 3.2 Data Stream Generators

This component aims at designing and implementing a specific DT for a specific scenario. The output of this component is a data stream simulating a real one but conditioned to a series of parameters and constraints. This component consists of three main modules, namely: (1) parameter setting, (2) a data generator composer, and (3) a data validator.

Parameter setting consists in defining the shape of the distributions we aim at for each of the entities involved in the DT. For example, we can define the particular distribution of user profiles we want in the data stream, biasing towards journalists or professionals.

The generator composed will select the most appropriate methods to generate the intended data stream according to the knowledge expressed in the SLOD-DT subgraph. This part involves traditional distribution generators, multimedia content generators (e.g., text and images) as well as time series generators.

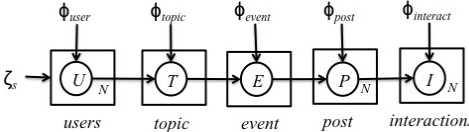

**Fig. 3.** Example of data generation for a social network DT.

Probability distributions are taken by default from SLOD-BI data but it can be changed in the parameter setting module.

Due to the complexity of a social network stream, designing a DT requires combining many different generative methods to simulate how users, topics, events, posts, and so on occur in that data stream. SLOD-BI patterns can help in defining the order in which different entities are generated and how they condition the other data to be generated. In Fig. 3 we show an example of a composed data generator following the boilerplate notation. Parameters are represented with $\phi_*$ and can be tuples of complex parameters. For example, $\phi_{post}$ is composed of the parameters for text, image, hashtag and mentions parameters. In our approach, we will regard both traditional probabilistic generation models and current state-of-the-art generators based on deep learning.

### 3.3    Data Stream Validator

As the data streams are randomly generated, we need to check that they do not contain inconsistencies, impossible values and incoherent contents. For this reason, we propose to apply consistency rules expressed in OWL2-RL to validate the generated data. Inconsistent data will be removed or replaced till the generated data stream becomes consistent and coherent. Examples of these constraints are: the limit values for user and post metrics, a user can only give a like or retweet once a post, and a user cannot interact to its own posts.

### 3.4    Analytical and predictive tools

There is a great variety of analytical tasks associated with social networks, for example: bot/spam detection, community discovery, user profiling, event detection, identifying influencers and checking data quality. Analytical tools aim at visualizing and detecting anomalies in data whereas predictive analytics are aimed at automatically classifying, predicting and recognising entities from data streams. Predictive analytics mainly rely on data-driven machine learning methods, which usually require many labeled examples. For both kinds of tools, the generation of simulated data is crucial for evaluating them in new scenarios before they are seen in real data streams. Dynamic SLOD-BI provides some of these tools which could be tested on the DTs outputs.

### 3.5   Use cases

A first scenario we want to address is that of Tourism. The intended DTs are mainly aimed at simulating the human behavior from a cognitive perspective. These DTs should be anthropomorphic representations of people who interact with tourist facilities and express their feelings about them. The internal structure of DTs will be designed by taking into account both cognitive and social aspects which must be also present in the KG. Currently, some cognitive data as well as image generation methods have been tested in this domain [18].

A second scenario is that of tracking fashion trends in social media. In this scenario we want to recreate the behaviour of *coolhunters* and the possible reaction of followers, for example to predict the stock of a new season after a new advertising campaign. The main idea is to develop a DT able to recreate new situations where image colors and text contents can be tuned according to some unseen trend. A preliminary work in this direction was presented in [19].

## 4   Conclusions

In this paper we propose a new paradigm for a semantic-driven definition of DTs for social networks. This proposal takes profit from the summarized data gathered from a Business Intelligence data infrastructure to set the parameters of a DT following the analyst requirements.

Semantic Web technology plays a relevant role in this approach since the SLOD-BI data and DT parameters and constraints are expressed according to the provided vocabularies and ontologies. Moreover, the approach relies on a dynamic KG representation since social data are continuously changing. Data generation algorithms are also expressed in the KG and linked to the concepts and parameters that best suit them. In this way, the definition and implementation of a DT is fully driven by the KG.

We plan to apply this proposal to some verticals already explored by the authors within the SLOD-BI project (e.g., automotive and medicine), as well new ones like Tourism and Fashion that would greatly benefit from social networks DTs. Preliminary results are expected soon for those domains where a good volume of data have been already gathered.

## Acknowledgments

This project has been funded by the Ministry of Economy and Commerce with project contract TIN2016-88835-RET and by the Universitat Jaume I with project contract UJI-B2020-15.

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
