# OpenReview forum: "Towards Semantic DigitalTwins for Social Networks"
_eswc-conferences.org/ESWC/2021/Workshop/SeDiT — SeDiT 2021 Oral_

### Official Review · AnonReviewer2 · 2021-03-24
**Ok idea, but the paper does not actually propose a digital twin**

**Rating:** 3
**Confidence:** 4

**Review:**

The paper proposes the concept of digital twins for social networks for use in analysis of societal reactions to trends and products/services. The twin would be based on semantic data from the SLOD-BI infrastructure, which was designed for this type of data, and modelled by combining this data with generative models. This setup makes sense, but it implies solving a lot of very hard problems. Generating representative and sane social media posts including the text and images is far from trivial and may not be possible (at the moment). I do however understand that this is a position paper and may be forgiven for not having all the answers.

The key problem is that a model of a social network does not qualify as a digital twin (DT) as I understand the term. The paper authors even include the definition of a DT at the start of section 2.1 as a: "digital representation, augmentation, and intelligent companion of its physical twin as a whole, including its subsystems and across all of its life cycles and evolution phases". Social networks are not physical systems and therefore their models are not digital twins in the sense that they are not digital representations of something physical. The proposed model could be very useful indeed, but simply does not match the definition of the term "digital twin".

There are some other issues that could be improved. I list them below, but the core problem is the definition issue.

Other issues:
 * Section 2.3 defines generative models as unsupervised learning methods. This is incorrect - generative models can be supervised. This part should be rewritten as the correct definition of generative models is completely in line with the rest of the text.
 * Section 3.3 proposes a data validator component which would validate the generated social network content based on consistency rules. The example rules proposed are fine and useful, but cannot ensure sanity of the content, which is very hard. For example vacation posts from London mentioning visits to the Eiffel Tower. Would results still be useful if such sanity checks are not performed?
 * The first use case proposes using this model to simulate human reactions to tourism facilities. It does not clarify if the reactions are to actual existing facilities or just artificial examples. If actual facilities are meant, then how are the parameters and properties going to be presented to the model for reaction. E.g. how is the food from a restaurant going to be represented so that the simulated reactions make sense? If artificial examples are meant, how are the results applicable to anything in practice? How are existing tourist facilities to learn something from simulated reactions to non-existing facilities? If the simulated reactions praise the taste of food, how is a real restaurant supposed to recreate that taste in reality?

 Overall, I think there is a key mismatch in the idea and the workshop topic and propose the paper authors seek publication elsewhere or re-work the idea so it matches the definition of a digital twin.

---

### Official Review · AnonReviewer3 · 2021-03-28
**Valid architecture, does not need Digital Twins**

**Rating:** 5
**Confidence:** 4

**Review:**

This paper describes a platform for the creation of Digital Twins for social networks. Authors discuss related work and motivate their work. An overview of the platform is described.

Given the fact that this is a position paper, a high level overview of the proposed architecture could be sufficient for a publication.

However, the proposed architecture could work without ever mentioning Digital Twins. Essentially, the part of the architecture that inputs a Knowledge Graph and outputs a data stream that adheres to specific parameters is labelled as a Digital Twin. Section 3.2 could be renamed to 'Data Stream Generator'.

---

### Official Review · AnonReviewer1 · 2021-03-31
**OK as a position paper but DT is not really necessary**

**Rating:** 4
**Confidence:** 4

**Review:**

The paper presents a system to simulate human activity on a social media platform.

The structure is fine for a position paper. Related work, underlying technology and a high level view of the system are presented together with a couple of potential use cases.

There is no real need to introduce digital twins to the system. These are more focussed on providing a digital representation of a physical device, object or system and as such are not really relevant for providing a simulation of users generating a data stream of interactions with a social network.

---

### Decision · Program_Chairs · 2021-04-08

Accept (Oral)